# Healthcare Recommender System Based on Medical Specialties, Patient Profiles, and Geospatial Information

Miguel Torres-Ruiz [1], Rolando Quintero [1,*], Giovanni Guzman [1] and Kwok Tai Chui [2]

1    Centro de Investigación en Computación (CIC), Instituto Politécnico Nacional (IPN),
     Mexico City 07738, Mexico
2    Department of Electronic Engineering and Computer Science, School of Science and Technology,
     Hong Kong Metropolitan University, Hong Kong, China
*    Correspondence: rquintero@ipn.mx; Tel.: +52-(55)5729-6000 (ext. 56577)

**Abstract:** The global outburst of COVID-19 introduced severe issues concerning the capacity and adoption of healthcare systems and how vulnerable citizen classes might be affected. The pandemic generated the most remarkable transformation of health services, appropriating the increase in new information and communication technologies to bring sustainability to health services. This paper proposes a novel, methodological, and collaborative approach based on patient-centered technology, which consists of a recommender system architecture to assist the health service level according to medical specialties. The system provides recommendations according to the user profile of the citizens and a ranked list of medical facilities. Thus, we propose a health attention factor to semantically compute the similarity between medical specialties and offer medical centers with response capacity, health service type, and close user geographic location. Thus, considering the challenges described in the state-of-the-art, this approach tackles issues related to recommenders in mobile devices and the diversity of items in the healthcare domain, incorporating semantic and geospatial processing. The recommender system was tested in diverse districts of Mexico City, and the spatial visualization of the medical facilities filtering by the recommendations is displayed in a Web-GIS application.

**Keywords:** recommender system; health attention factor algorithm; application ontology; semantic similarity; Web-GIS application

## 1. Introduction

The global outburst of the new coronavirus (SARS-CoV-2) has introduced severe concerns regarding the capacity and adaption of healthcare systems and specifically about how vulnerable citizen classes might be affected [1]. The COVID-19 pandemic has generated the most significant transformation and disruption of many services, including healthcare and health emergency management, using digital tools, which have increased rapidly to bring sustainability to health services. Moazzami et al. [2] and Katz et al. [3] assert that a pandemic seriously alters global health systems in transmission, control, and saturation but also restricts resources and facilities, medical personnel, vaccines, access, and mobility issues.

Thus, digital technologies such as artificial intelligence, blockchain, machine learning, the Internet of Things, and extensive health data repositories have contributed to changing and addressing the traditional methods of providing health services to citizens and innovating in different paradigms, such as mobility, from a disease perspective to patients' perspective, well-being citizens and life quality. Moreover, the digital transformation focused on healthcare is continuously impacting medical approaches [4,5]. Nowadays, this transformation and innovation implicate new stakeholders conducted by massive patient data from various datasets in different formats, computational intelligence models and algorithms, artificial intelligence-based techniques, novel pervasive platforms for exchanging and monitoring patients, and service providers for obtaining valuable information, patterns, trends and insights in the healthcare domain.

According to Kraus et al. [6], there are five research classifications concerning the digital transformation of healthcare: (1) patient-centered technology, (2) operational efficiency of organizations, (3) managerial implications, (4) impact on workforce practice, and (5) socioeconomic aspects. By taking into consideration this categorization, sustainability in healthcare should cover all these issues to adopt new strategies, methodologies, and innovative applications to provide health value services-based technologies and responses to face future global health emergencies.

Since the worldwide outbreak and lockdown restrictions of the COVID-19 pandemic, diverse initiatives to maintain the activity of health services have been implemented. For instance, virtual medical consultations have increased in many countries [7], and some medical departments continue using online technology to monitor non-critical patients [8,9]. According to Shahed et al. [10], the pandemic has disturbed the supply chains around the world, generating issues related to medical response support, lack of essential medical supplies, deficiency in access and mobility to clinical and hospitals to treat medical specialties, and the resilience of citizens. Indeed, 94% of 1000 Fortune medical and supplier companies are suffering interruptions in their supply chains, 75% of these companies have experienced a negative impact on their business operations, and 55% of these companies are scheduling to reduce their growth plans.

Thus, Rosenbaum [11] presented a study that asserted the pandemic transformed the use of health services to attend to different emergency conditions in patients without COVID-19, showing a general reduction in the treatment of medical specialties. The study suggests that this phenomenon is due to (1) negligence by patients with severe or life-threatening disorders to pursue care, (2) release of the emergency units for non-emergency disorders, or (3) displacement of emergency unit care to other locations, such as telemedicine consultations. Moreover, in an analytical study reported in the United States, the visits and admissions to emergency units decreased by more than 40% in all the healthcare systems. This fact was probably generated by the citizens' response to the sanitary emergency messaging concerning COVID-19 in the country. Thus, the population avoided assisting emergency units because of the fear of being exposed to the virus, particularly in these healthcare spaces.

According to Kumar and Sharma [12], the recommendation task is considered a classification and ranking prediction model. Consequently, there are different and widely used techniques to generate such recommendations. In this context, Lika et al. [13] categorized five approaches in the recommender systems: (1) Collaborative Filtering (CF) technique that recommends elements to the user considering previous characteristics of other users with similar conditions based on user profiles. (2) Content-based Filtering (CB) suggests elements that are similar to others based on a given previous user profile. (3) Social Filtering (SF) that recommends elements considering suggestions and preferences taken from social networks (particularly friends). (4) Knowledge-based System (KB) recommends elements to the users relating particular domain knowledge considering a similarity value to assess the matching grade; and (5) the Hybrid Technique (HB) combines two techniques to recommend elements to the user.

There are different research issues and challenges concerning those approaches to improve the recommendations in recommender systems. Thus, Kumar and Sharma [12] and Lika et al. [13] described a relevant research gap in implicit and explicit methods to collect ratings for the matrix, which reflects a *sparsity* problem in which there are no scores to know the satisfaction level of users. The *cold-start* problem is associated with missing information concerning offering recommendations to new users in the system because it does not have ratings to initiate the task. Moreover, the *scalability* issue is a big challenge because it is difficult to find open architectures and algorithms to increase the functionality of the recommender systems and handle dynamic and large datasets or preference collections. Concerning the *privacy and robustness* problems is essential to guarantee the users that their preferences and confidential information are safe in the system. Thus, *cybersecurity* approaches are needed to ensure the data of users and the

confidence to access the recommender systems. In addition, *recommenders in mobile devices* are relevant to design a new generation of RS focused on location-based services. Nowadays, geographic information plays an important role in generating recommendations for many services, employing social media information to analyze preferences and tastes. Thus, it demands more computational solutions concerning effective mobile user interfaces and efficient algorithms to retrieve information on the preferences and tastes of users. Finally, the *diversity of items* implies the inference and knowledge discovery based on novel techniques to provide particular recommendations, considering a broad scenario of preferences. To date, there is little research related to this issue to improve the accuracy of the recommendations.

Considering the challenges of the state-of-the-art, we propose an approach centered to tackle issues related to recommenders in mobile devices and the diversity of items in the healthcare domain. Thus, we focused on incorporating semantic and geospatial processing in a particular RS as well as an attention factor index to generate recommendations to users based on medical specialties and their nearby facilities.

Therefore, we perceive the following issues concerning medical attention: less attention has been focused on geographic access to health services, hospital access, and response capacity at the local level within urban areas, and lack of access to medical specialties according to the responsiveness and infrastructure. Thus, we propose a novel approach based on patient-centered technology. In this way, we have designed a recommender system that consists of a health service level, which is defined by a health attention factor. This metric is composed of two key components, the geospatial location of the health facilities and the medical specialties required by the patients.

The main contribution of this research work is related to the health attention factor that takes into consideration different criteria to compute the recommendation list based on the following: user-required specialty, current location, arrival time from some volunteer-collaborative service (Google Maps or Wazee), medical facility, occupation factor, number of medical doctors classified by specialty, and similarity metric. Finally, this is a customizable approach because some criteria could be modified to meet other requirements, such as buffer distance and similarity metric, among others.

The rest of the paper is structured as follows: the next section comprises the literature review on intelligent systems and approaches focused on health services and their implications during the COVID-19 pandemic, Section 3 describes the methodology, Section 4 investigates the results, and the last section discusses the key findings of our research.

## 2. Related Work

Systems based on Artificial Intelligence (AI) have permeated many areas of life. Thus, healthcare is not an exception; multiple approaches have supported health services with this type of tool. There are many challenges to achieving the aforementioned, such as investigating biases in clinical decision-making and lack of trust in AI by people unaffiliated with these technologies, among others.

The study conducted by Lai et al. [14] identified gaps and proposed future research directions. The authors reported that there are limited studies concerning the interactive collaboration task in healthcare and that there is no good integration between people and AI.

On the other hand, with the health crisis that has tied the world in recent years, telemedicine services have been essential for many users to access health services beyond this required assistance by the pandemic emergency related to COVID-19. Chauhan et al. [15] identified fundamental success factors relevant to telemedicine services and grouped them under some contextual criteria. The findings revealed that the more complicated the technology is, the more resistance there is to adopting it. Baudier et al. [16] examined the factors to predict the intention to use medical teleconsultations, and the results highlight the importance of trust beliefs and self-efficacy in digital health services adoption.

Therefore, in this context, it is necessary to understand what is required of a new type of company that provides health services using new remote work technologies. Chakraborty et al. [17] attempt to understand the status of these health technology companies in providing healthcare services through a study of scientific publications on the matter. With a total of 110 journals reviewed, 76 articles were found to meet the inclusion criteria, and only five studies portrayed the status of new health technology companies in the provision of healthcare services. Similar results are presented by Qahtan et al. [18]; the study concluded that despite the efforts to develop safer and more private systems for the health industry, none meets the necessary development attributes.

Another critical aspect to consider is that access to these services must be viable and available to all people regardless of their socioeconomic status. For this reason, there must be criteria for evaluating health technologies (HTA). According to Drummond et al. [19], there are a few challenges to such an evaluation, taking into account that resources are limited to run HTAs locally. Another problem is the low availability of domestic data to complete the profitable models and the timely readiness of pertinent HTAs, among others. Many Latin American countries have parallel health systems in which mandatory health insurance or social security systems for workers, subsidized public programs, and private mechanisms coexist. Meleddu et al. [20] asserted that research has shown that people make health spending decisions based on their income, political ideology, and demographics, and there is a responsibility to notice public healthcare is a normal commodity that is barely available. Thus, users demand health services that meet their expectations.

In general, we consider that patients can have access to public health services, and some other patients have access to private health services that are more expensive. Therefore, public services often cannot provide health services remotely, making it necessary for people to travel efficiently to the most convenient health centers. Mollahaliloglu et al. [21] mention that from the years 2002 to 2016, policies of redistribution of health services were applied, resulting in a continuous decrease in inequalities in the geographical distribution of the human workforce in the health sector. Thus, applying policies that impact the quality of health services is critical. For example, Sharma and Patil [22] present the accessibility measure for health services by using public transport, the travel time, and the number of transit stops. In addition, Pereira et al. [23] show how transport-accessibility analytics can provide actionable insights to improve healthcare coverage and responsiveness.

Medical professionals have a lot of medical information, which has made it difficult to make patient-oriented decisions. Recommender Systems (RS) can help in making such decisions so that they are more accurate. According to Tran et al. [24], three main aspects must be considered in RS: (a) the context of use, (b) users are the final consumers of RS, and (c) the elements are the inputs that users are looking for. The authors also mention that there are four basic recommendation techniques: (1) Collaborative Filtering (CF) recommendations consider that a patient who has similar health conditions (profiles) to other patients should have similar health care treatments to the latter. (2) Content-Based filtering approach (CB) recommends appropriate health services to the patient's health situation and similar to those previously assigned. (3) Knowledge-Based Recommendation approach (KBR) generates recommendations founded on knowledge about commodities, explicit user preferences, and constraints that express dependencies between user preferences and object properties. (4) Hybrid Recommendation approach (HR) tries to mix the aforementioned recommendation strategies to take advantage of one technique and correct the weaknesses of another. Pincay et al. [25] present the results of a comprehensive and cutting-edge study of RS used in the health care context, also known as health recommender systems.

Recently, multiple approaches have been generated to solve this RS problem. Shaikh et al. [26] proposed a framework for an RS for dengue patients in healthcare applications. In this framework, machine learning algorithms are used, especially content-based, collaborative and hybrid approaches. In the same vein, Vairale and Shukla [27] analyzed recent research in the field of a healthy lifestyle, in which individualized recommendations are made based on clinical data. Fasidi and Adebayo [28] presented the rule-based naive

Bayesian classifier (RNBC) as a prediction model for heart risk conditions and a therapy handbook. Gohari et al. [29] introduced the significance-based confidence-aware recommendation method (SBTAR), which operates a trust metric that employs the commodity importance paradigm. Sahoo et al. [30] mentioned that health RS can be used to obtain additional information concerning a person's medical care. Such systems identify preferred hospitals by computing the similarity between choices made by patients. Waqar et al. [31] proposed an adaptive algorithm for the effective generation of medical recommendations. The system could be improved by adding patients' treatments and symptoms of a particular illness. Mazeh and Shmueli [32] proposed a model for an RS based on the storage of patient data and focused on improving confidentiality while maintaining the reliability of the recommendation. In the model, collaborative filtering, personal data storage, and the content-based approach are used to preserve confidentiality. Sayeb et al. [33] have aimed to present a graph-based RS to manage the COVID-19 crisis considering data from patients and medical personnel. The RS initially analyzed the medical records of the patients to determine which profile of medical personnel may help a patient in a crisis situation. Thus, the RS will try to propose other doctors with the same profile and the closest competencies and abilities.

Similarly, Zhang et al. [34] proposed the iDoctor system to provide users with personalized medical recommendations. This application examines users' emotions and preferences regarding doctors by means of their ratings and reviews.

Community detection is a topic of relevance in social network analysis. Luo et al. [35] investigated the performance of community detectors based on non-negative symmetric matrix factorization (SNMF) with a scaling factor adjustment. Several SNMF schemes are improved by adjusting the scale factor in a non-negative multiplicative update (NMU) scheme in a linear or non-linear strategy, thereby introducing new community detectors. The results indicate that they outperform original SNMFs in predicting the potential community of unlabeled nodes.

Narducci et al. [36] presented a social network called HealthNet, where a recommendation component is integrated to suggest the doctors and hospitals that best fit a specific patient profile. Based on patient health data, the database is searched for patients with similar conditions. (Equation (1)).

$$s(a,b) = \alpha \frac{\sum_{c_a \in C_a, c_b \in C_b} s_c(c_a, c_b)}{|C_a| + |C_b|} + (1 - \alpha) \frac{\sum_{t_a \in T_a, t_b \in T_b} s_t(t_a, t_b)}{|T_a| + |T_b|} \tag{1}$$

where $a$ y $b$ are patients, $C_x$ is the set of conditions of the patient $x$, $T_x$ is the set of treatments for the patient $x$, $\alpha$ is a parameter for regulating the ratio between contributions of conditions and treatments. $s_c(c_a, c_b)$ is the similarity between two conditions: in the case that $c_a = c_b$ the similarity $s_c(c_a, c_b) = \log \frac{|C|}{|P_{c_a}|}$, being $C$ the universe of conditions and $P_{c_a}$ is the set of patients with condition $c_a$; in the case that $c_a \neq c_b$ the similarity $s_c(c_a, c_b) = \frac{1}{\delta(c_a, c_b)}$, being $\delta$ the length of the shortest path between $c_a$ y $c_b$ in the hierarchy of diseases. $s_t(t_a, t_b)$ is the similarity between two treatments and is equal to 1 if $t_a = t_b$, 0 otherwise.

Now, based on this similarity of patients, a rating is calculated for doctors (Equation (2)) and hospitals (Equation (3)) with respect to patient $p$.

$$scoreDoc(d, p) = \sum_{\rho \in P} s(p, \rho) \cdot r_\rho(d) \tag{2}$$

where $P$ is the set of registered patients, and $r_\rho(d)$ is the rating given by the patient $\rho$ to the doctor $d$.

$$scoreH(h, p) = \beta \left( \sum_{\rho \in P} s(p, \rho) \cdot r_\rho(h) \right) + (1 - \beta) \cdot Q(h) \tag{3}$$

where $r_\rho(h)$ is the rating given by the patient $\rho$ to the hospital $h$, and $Q(h)$ is a quality index of the hospital $h$ given by a rating authority.

## 3. The Proposed Approach

According to Neutens [37], the most common parameter for measuring geographic access to healthcare services is the shortest distance or travel time to arrive near a medical unit. This metric is widely used and easy to compute and analyze and, therefore, clear to convey to policymakers. However, the most critical limitation of this metric is that it discharges the traffic congestion and the implicit events that originate from this condition. There are other potential approaches, such as the Floating Catchment Area (FCA) methods [38]. Thus, Matthews et al. [38] computed the accessibility levels considering the following variables: the provider-to-population ratio of every health unit and the potential health demand regarding the infrastructure. Nevertheless, the limitation of this method concerns the overestimation of both health service demand and supply, which could induce untruthful accessibility estimation values.

Therefore, we propose a novel approach based on patient-centered technology according to the classification proposed by Kraus et al. [6]. In this way, we design a recommender system that consists of a health service level defined by a health attention factor. This metric is composed of two key components, the geospatial location of the health facilities and the medical specialties required by the patients.

The approach consists of a recommender system of health services considering the geographic location, the medical specialties defined within an application ontology, and the attention factor to offer the best health unit to treat a medical situation. Therefore, when a medical condition is presented, the patient requires assistance at a medical facility for partial or complete medical attention. In this case, two scenarios could be presented: (1) the patient can readily identify the adequate medical unit by making a query in the recommender system, and it returns a set of medical centers to the patient according to geographic location criteria. (2) However, there is no *a priori* information about the most acceptable hospital for attending the emergency patient. It is possible that the nearest hospital does not have a doctor with a neurology specialty; so, according to this medical situation, a recommender system semantically processes the query to provide information concerning this health service. Therefore, the recommender system performs the following duties: semantic-based recommendations, profiling of answers, and map rendering. Figure 1 presents the processes of the recommender system to retrieve semantic medical information about the medical centers.

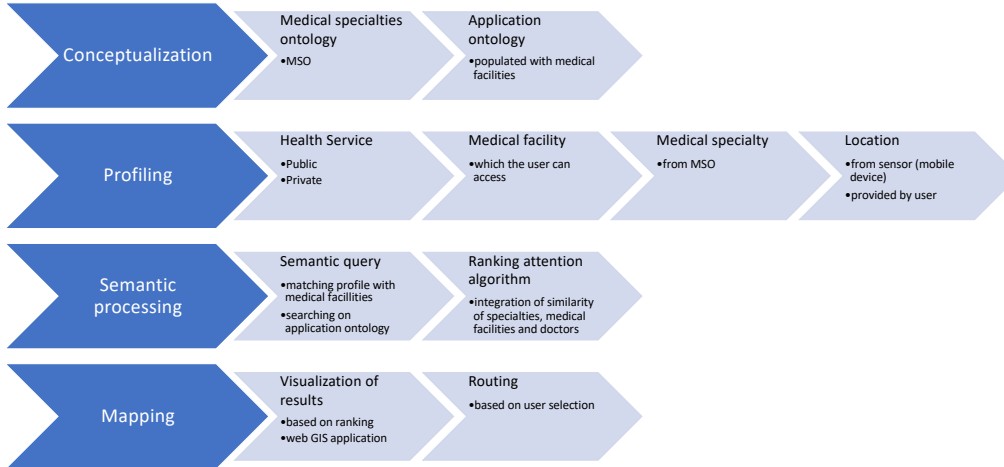

**Figure 1.** Processes of the proposed recommender system.

### 3.1. The Medical Application Ontology

The medical application ontology was designed for semantic processing. This ontology conceptualizes the most essential medical specialties considering the Association of American Medical Colleges Standard (https://www.aamc.org/cim/). In this context, other

medical ontologies were created according to their applications such as SNOMED [39], MedO [40], OBI Ontology [41], GALEN [42], Gene Ontology [43], among others.

Those ontological representations are built with diverse computational environments such as Resource Description Framework (RDF), Ontology Web Language (OWL), and Open Biomedical Ontology (OBO); all of them based on a symbolic language. The structures and syntaxis to describe entities and relationships are very complex, and they contain up to 900,000 concepts in their topological structure. These features are difficult and slow to process semantically, and the medical specialties are not explicitly defined in such ontologies. Moreover, the interoperability of these public ontologies is a big challenge because, in production systems, the performance can decrease significantly due to several issues, such as diverse access to different ontologies, translation among languages, different reasoning, and inference engines to provide semantic information from queries, and semantic information integration techniques.

Thus, the proposed research proposes an application ontology that consists of a hierarchical structure based on an is-a relationship, with partitions associated with medical specialties. Summing up, the proposed ontology fits with the recommendations generated by the system with a conceptualization more suitable for querying medical specialties due to a single hierarchical description, specialized partitions, and granularity. We develop the proposed ontology with the OWL-full language using the Protégé application with version 5.5. The root node of the ontology has three properties to perform the semantic processing: the first one establishes if the medical specialty is "diagnostic" or "therapeutic". The second property specifies the patient age defined by ("pediatric", "adult", "geriatric", "all"), and the third depicts if the medical specialty is "organ-based", "technique-based" or "both". Figure 2 shows the application ontology to conceptualize medical specialties.

### 3.2. The Profiling and Semantic Processing Tasks

The profiling task establishes a user personalization (profile) and the statement of the search (query type). Therefore, the profile for each query is composed of the following features:

- Health Service Type. The user defines if he demands private or public health services.
- Medical Facilities. All units, centers, and hospitals that accomplish the health service type are shown.
- Medical Specialties. They must identify the disciplines for each previously enlisted institution or medical center.

The semantic processing task is in charge of receiving the search type as an input parameter. The value of the search type defines the query to build in the application ontology using the SPARQL language. In this way, the semantic information concerning the medical centers, units, or hospitals is retrieved by taking into consideration the statements of the user.

Moreover, the type assumes one of the following potential values: general, emergency, or profile-oriented. Regarding the general type of search, the retrieved list of medical facilities is ranked taking into account the geospatial distances that were measured according to the user's location. In conclusion, the first medical facility is the nearest user's patient. Therefore, in an emergency case, the "ER" string is allocated to the variable of search type, and the preceding operations are made, but the outcome just has medical facilities that supply emergency services. In a profile-oriented query, all the information provided in the user's account is used to find the closest medical facilities that offer the medical specializations mentioned in the account. This task will produce a collection of hospitals represented by the set $MF = \{mf_1, mf_2, \ldots, mf_n\}$, where $MF$ is the set of medical facilities, centers, or hospitals.

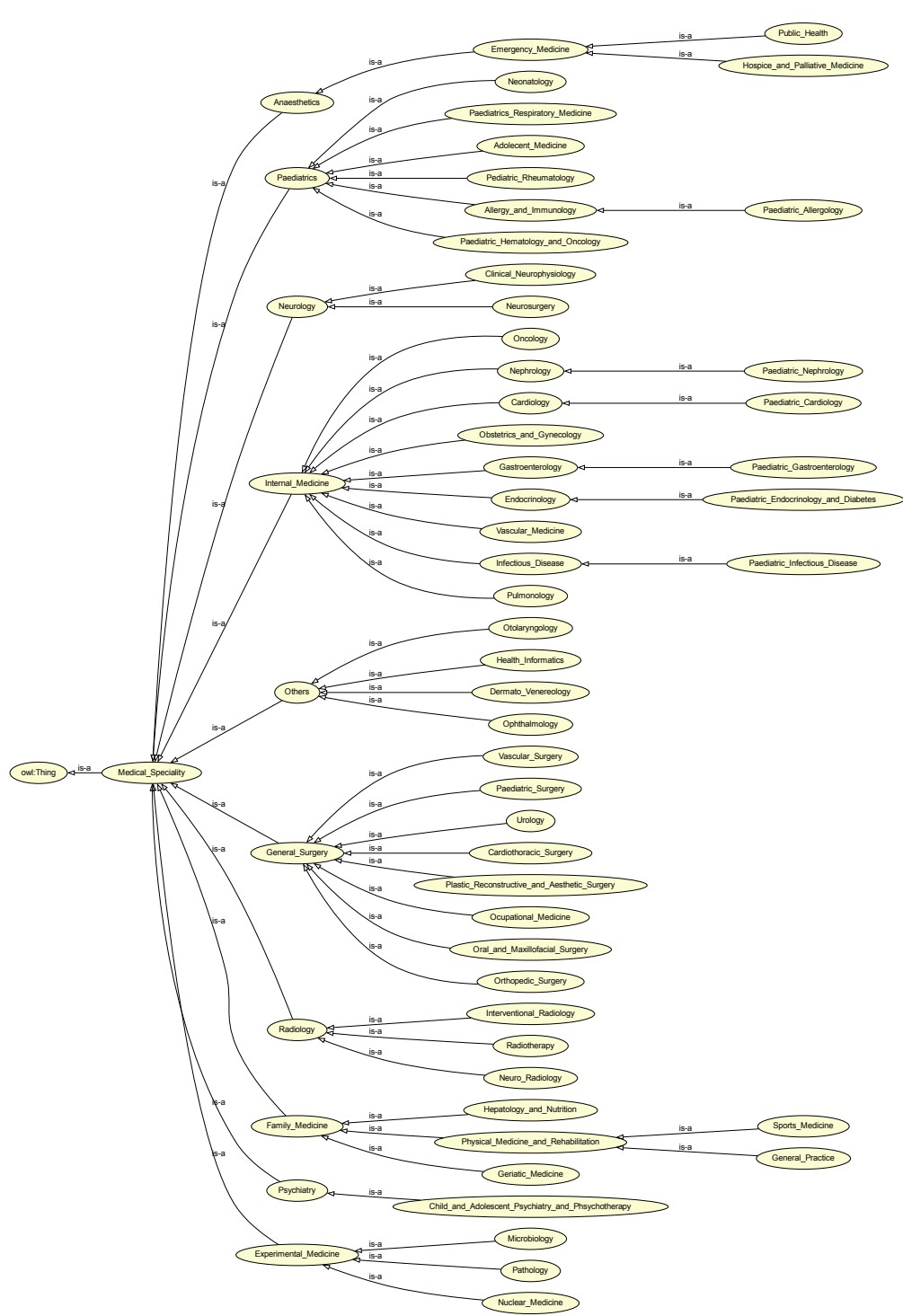

**Figure 2.** Application ontology representing medical specialties.

## 3.3. The Ranking Attention Factor

We propose a novel ranking algorithm for determining the health attention factor and sorting the medical facilities that accomplish a medical specialty. Therefore, the algorithm computes the attention feature for a collection of medical units, centers, or hospitals numerically. For example, in an emergency such as a heart attack, it is desirable for a doctor with a cardiology specialty to care for the patient. Nevertheless, in the case that this doctor or specialist is unavailable, it is critical to calculate the semantic similitude and identify all potential physicians with some corresponding knowledge competencies to treat the patient.

Consequently, we design the following equation to determine the number of doctors who can respond to an emergency involving a particular medical discipline. This value specifies the medical specialties semantically associated with the desired specialization, as indicated in the user profile. Let $f$ be a function where $MF_B$ is the number of medical facilities within a ratio obtained by using the geographic buffer operator fro the current user geographic location, **S** is the collection of all areas of medicine specified by the application ontology, and $\mathbf{S}_{mf_i}$ is a subgroup that indicates the amount of clinical specialties provided by a particular medical facility. Equation (4) is then utilized to extract the number of medical doctors for a certain hospital.

$$related\_doctors(mf_i) = \sum_{\forall s \in \mathbf{S}} medical\_doctors(s, mf_i) * sim(s, s_u), mf_i \in MF_B \qquad (4)$$

where $medical\_doctors(s, mf_i)$ represents the number of health practitioners (doctors) with a specialty $s$ in the $mf_i$ medical facility, and $sim(s, s_u)$ calculates the semantic similarity between $s$ and $s_u$ according to the user profile described by $u$. Thus, to compute this similitude value, the semantic similarity proposed by Resnik [44] between two terms $t_1, t_2$ is applied. In this case, we use Equation (5).

$$sim(t_1, t_2) = \max_{t \in S(t_1, t_2)} [-\log p(t)] \qquad (5)$$

where $S(t_1, t_2)$ is the collection of common ancestors of two terms $t_1$ and $t_2$. The Resnik similarity has a minimum of zero.

### 3.4. The Map Rendering Task

This work utilizes Google Maps API version 2 to interpret the ranking algorithm's output and represent the healthcare centers on a map. In contrast to other map servers, such as Yahoo Maps and Microsoft Bing Maps, among others, this map server provides superior download rate, display, and response times for mobile devices. Moreover, we use a temporal tile to add further visual analysis possibilities (markers, polygons, route lines). This activity analyzes the incoming data containing a geographically ordered (generic and oriented-profile queries) or sorted (urgent situation query) collection of institutions to count markers for all medical facilities included in the input list. Therefore, a geographic marker represents each medical facility and a label containing the facility's name and address. In addition, the distance is measured in kilometers. Therefore, to obtain an estimation of the time required to arrive at the desired location, we operate with Google Directions API. The developed query has the following pattern:

https://maps.googleapis.com/maps/api/directions/json?
origin=*latitude_origin*, *longitude_origin*&
destination=*latitude_destination*, *longitude_destination*&
key=*YOUR_API_KEY*

On the other hand, it is possible to compute the geographic distance between the two locations (origin and destination), applying the Haversine function [45]. See Equations (6)–(8).

$$distance(o(lat_1, lon_1), d(lat_2, lon_2)) = R \times c \qquad (6)$$

$$a = sin^2\left(\frac{rad(\Delta_{lat})}{2}\right) + cos(rad(x_1)) \times cos(rad(x_2)) \times sin^2\left(\frac{rad(\Delta_{lon})}{2}\right) \qquad (7)$$

$$c = deg\left(2 \times arcsin\left(min(1, \sqrt{a})\right)\right) \qquad (8)$$

where:
$\Delta_{lon} = lon_2 - lon_1$ is the longitude coordinates difference.
$\Delta_{lat} = lat_2 - lat_1$ is the latitude coordinates difference.
$rad(value)$ is a function to transform *value* from degrees to radians.
$deg(value)$ is a function to transform *value* from radians to degrees.

$R$ equal to 6378.137 kilometers is the Earth's radius.

Wu et al. [46] proposed an $L^3F$ model to deal efficiently with the high-dimensional and sparse (HiDS) matrices coming from the recommender system. This model is based on the combination of the $L_1$ and $L_2$ norms to calculate its loss function. In said combination, the weights of the $L_1$ and $L_2$ norms are adaptively adjusted [47,48]. Experiments showed that the model increases in robustness (due to the $L_1$ norm) and stability (due to the $L_2$ norm) when dealing with a HiDS matrix with outliers. Thus, we used a loss function based on $L_1$.

On the other hand, the complete pseudo-code of this proposal is described in Algorithm 1. This algorithm describes all the steps required to compute the adequate medical unit by combining the arrival times of all hospitals and the estimation of the attention factor; thus, we used a loss function based on $L_1$. In addition, this approach considers the medical specialty required by the mobile application user, their current location, and the specialties and medical doctors in each medical facility. Once the ranked list has been determined, the directions list must be requested to travel from the current user's location to the high-ranked medical facility using Google directions. Then, from the response body, it is possible to obtain the point list to reach the target hospital.

---

**Algorithm 1:** Algorithm to generate the recommender list of medical facilities

---

**Input:**

$MF_B = \{mf_1, mf_2, ..., mf_n\}$, set of medical facilities inside buffer B.

$s_u$, specialty required by the user.

$R$, search ratio to apply buffer operator.

$pat(mf_i) \ \forall \ mf_i \in MF$, current patients of $mf_i$.

$cap(mf_i) \ \forall \ mf_i \in MF$, maximum patients capacity of $mf_i$.

$P_u(lon_u, lat_u) \leftarrow$ user current latitude and longitude.

$max(at(P_u, P_{mf_k})) \leftarrow$ maximum arrival time.

**Output:**

$Route(P_u, AF_B(0)) = \{p_1, p_2, ..., p_n\}$

**for** $i = 1 \to n$ **do**

   $P_{mf_i}(lon_{mf_i}, lat_{mf_i}) \leftarrow i-$medical facility latitude and longitude.

   $|at(P_u, P_{mf_i})|$

   $related\_doctors(mf_i) = 0$

   **for** $j = 1 \to m \ \forall \ s_j \in S_{mf_i}$ **do**

      $related\_doctors(mf_i) + = medical\_doctors(s_j, mf_i) * sim(s_j, s_u)$

   **end**

   $af(mf_i) = related\_doctors(mf_i) * [1 - pat(mf_i)/cap(mf_i)]$

   $af(mf_i) = af(mf_i) * [1 - at(P_u, mf_i)/max(at(P_u, P_{mf_k})]$

**end**

$AF_B = \{mf_1, mf_2, ..., mf_n\}, af(mf_i) > af(mf_k), i < j \le n.$

$P_{origin} = P_u(lon_u, lat_u).$

$P_{destination} = AF_B(0) \leftarrow$ medical unit at index zero.

$base\_url = https://maps.googleapis.com/maps/api/directions/json?$

  $response = base\_url + origin = P_{origin}\&destination = P_{destination}\&key = $

  $API\_KEY.$

$Route(P_u, AF_B(0)) = response.routes.steps$

---

## 4. Results and Discussion

This section describes a comparative result of the attention factor algorithm, with a set of four medical facilities computed after applying the buffer operator, at a radius of 1000 m, from the user's geographic current position to the clinical units. Table 1 describes

the medical specialties and the algorithm parameters, such as the number of patients and the maximum capacity of each medical center.

**Table 1.** Information on specialties and patient capacity of a group of hospitals.

| Hospital | Patients | Capacity | Oncology | Cardiology | Obstetrics and Gynecology | Emergency Medicine | Nephrology |
|---|---|---|---|---|---|---|---|
| $mf_1$ | 15 | 50 | 5 | 1 | 3 | 2 | 3 |
| $mf_2$ | 10 | 30 | 2 | 2 | 0 | 3 | 4 |
| $mf_3$ | 60 | 80 | 6 | 2 | 5 | 3 | 2 |
| $mf_4$ | 50 | 120 | 6 | 3 | 4 | 5 | 4 |

We observe that in each medical facility, medical doctors specialize in one of the following disciplines: oncology, cardiology, obstetrics and gynecology, emergency medicine, and nephrology. The occupation percentage ranges from 30% (for $mf_1$) to 75% ($mf_3$). According to these criteria, the sorting process to select the hospital is $mf_1, mf_2, mf_4$, and $mf_3$. Taking capacity into account, the following should be the preferences for attending the medical facility: $mf_4, mf_3, mf_1$, and $mf_2$.

On the other hand, Table 2 shows the values obtained by applying the Resnik similarity measure, as well as the attention factor, arrival time, and normalized values described in the previous section.

**Table 2.** The obtained values by applying the attention factor algorithm.

| Hospital | Arrival Time (Minutes) | Related Doctors | Attention Factor | Final Attention Factor |
|---|---|---|---|---|
| $mf_1$ | 12 | 8.1859241 | 5.7301469 | 1.1460293 |
| $mf_2$ | 10 | 6.6240311 | 4.4160207 | 1.4720069 |
| $mf_3$ | 15 | 12.373001 | 3.0932502 | 0.0000000 |
| $mf_4$ | 13 | 14.466539 | 8.4388146 | 1.1251752 |

According to the results presented in Table 2, the medical facility $mf_3$ contains the best care factor, considering the cardiology specialty required by the user created in this scenario. These results were obtained by combining the arrival time (in this example from Google Maps), the number of medical doctors, their specialties, and the semantic similarity by applying a similarity metric. In a parallel process, as explained in Table 1, the preferences changed in one-criteria selection. For instance, based on arrival time, the best option is $mf_2$, followed by hospitals $mf_1$, $mf_4$, and $mf_3$. By using the specialties and similarity metric, the recommendation list is $mf_2$, $mf_1$, $mf_3$, and $mf_4$ (the last two items changed). Now, by computing the attention factor and determining the normalized scalar value, the best option is $mf_3$ (the recommendation list is $mf_3$, $mf_4$, $mf_1$, and $mf_2$.

Figure 3 shows an extract of the relevant information from the response file in JSON format, which is obtained using the Google Directions API, taking as parameters for the query: the origin with coordinates (19.503147, −99.147667), and being a medical facility with coordinates (19.4663758, −99.147163). This response extracts descriptive information with indications of the path to follow to reach the point of interest.

To calculate the estimated time for arrival, the user can select one of the following options: 'driving', 'walking', and 'bicycle'. Figures 4 and 5 depict the result of computing the optimal route.

A comparative result is described in Table 3, which contains the arrival time obtained (as a manual process) using the Google Maps and Waze applications. Unfortunately, these applications do not publish the collaborative information obtained from users, and we can only obtain the arrival time at the moment of the query. According to Google Maps, applying an ascending sorting process and taking into account the arrival time, the medical facilities are $mf_2, mf_1, mf_4$, and $mf_3$. On the other hand, to obtain the same information

in Waze, the sorting process is $mf_2, mf_3, mf_1$, and $mf_4$. Finally, computing the proposed normalized attention factor, the result is $mf_3, mf_4, mf_1$, and $mf_2$.

```
1  {
2    "geocoded_waypoints": [
3      {
4        "geocoder_status": "OK",
5        "place_id": "ChIJPRjznEr40YUR4bvV0G3n73g",
6        "types": [
7          "establishment",
8          "point_of_interest",
9          "university"
10       ]
11     },
12     {
13       "geocoder_status": "OK",
14       "place_id": "ChIJhQcO7Bz50YURys8p0rKRjAg",
15       "types": [
16         "street_address"
17       ]
18     }
19   ],
20   "routes": [
21     {
22       "bounds": {++},
23       "copyrights": "Map data  2022  INEGI",
24       "legs": [
25         {
26           "distance": {++},
27           "duration": {
28             "text": "10 min",
29             "value": 611
30           },
31           "end_address": "Av. Insurgentes Nte. 848, Vallejo Poniente, Gustavo A. Madero,
                  07790 Ciudad de M xico , CDMX, M xico",
32           "end_location": {++},
33           "start_address": "Av. Juan de Dios B tiz S/N, Nueva Industrial Vallejo,
                  Gustavo A. Madero, 07738 Ciudad de M xico , CDMX, M xico",
34           "start_location": {++},
35           "steps": [++],
36           "traffic_speed_entry": [++],
37           "via_waypoint": [++]
38         }
39       ],
40       "overview_polyline": {++},
41       "summary": "Eje Central L zaro C rdenas",
42       "warnings": [++],
43       "waypoint_order": [++]
44     }
45   ],
46   "status": "OK"
47 }
```

**Figure 3.** Query result in JSON format obtained using Google Directions API.

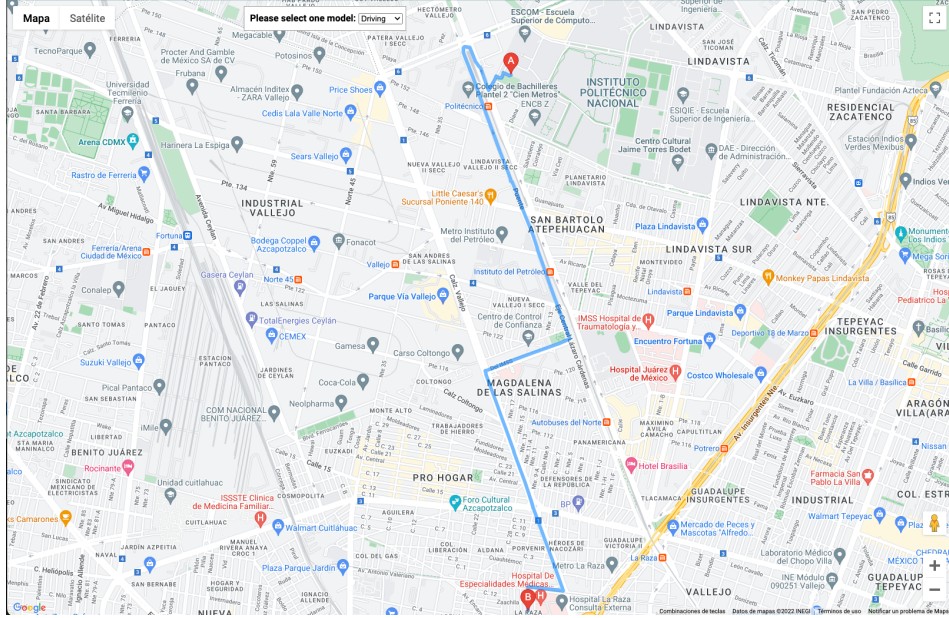

**Figure 4.** Visualization of the route. Case 1.

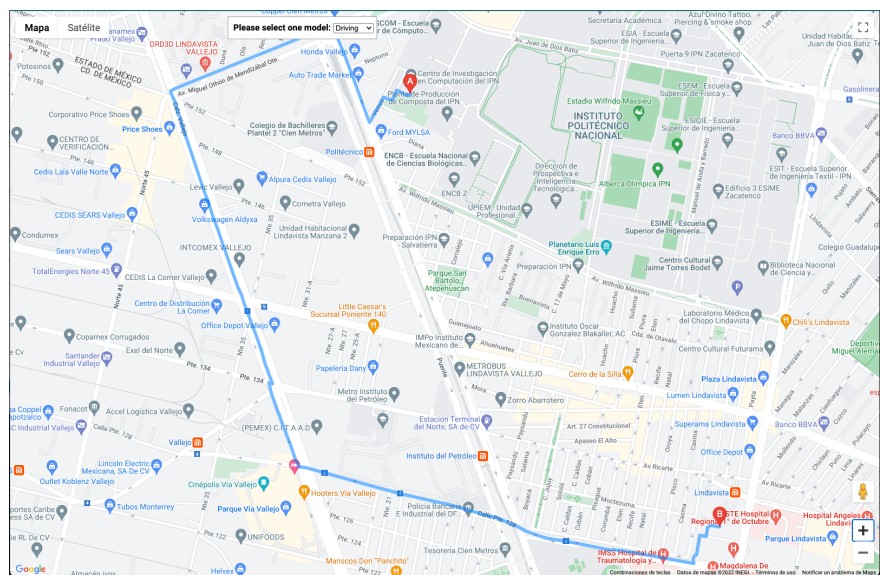

**Figure 5.** Visualization of the route. Case 2.

**Table 3.** Comparative process using Google Maps, Waze App, and the proposed method.

| Hospital | Arrival Time (Google Maps) | Ranking | Arrival Time (Wazee) | Ranking | Proposed Algorithm | Ranking |
|---|---|---|---|---|---|---|
| $mf_1$ | 12 | 2 | 13 | 3 | 1.1460293 | 3 |
| $mf_2$ | 10 | 1 | 11 | 1 | 1.4720069 | 4 |
| $mf_3$ | 15 | 4 | 12 | 2 | 0.0000000 | 1 |
| $mf_4$ | 13 | 3 | 14 | 4 | 1.1251752 | 2 |

In this sense, we assume that the responsibility of clinicians and public health officials is to emphasize to patients the significance of ongoing attendance to medical centers, units, or hospitals to receive appropriate treatments, not only in emergencies but also for continuous and progressive actions to monitor diseases. Nevertheless, patient-centered technology has demonstrated an innovative performance in managing and monitoring clinical settings, providing sustainable actions for the well-being of citizens.

## 5. Conclusions and Future Work

In this paper, we propose a methodological and collaborative architecture to promote the assistance of citizens to medical facilities. The approach consists of a recommender system in which the health attention factor metric assesses which medical facility, center, unit, or hospital with better infrastructure and diverse health specialties is available to bring attention to patients. Thus, the proposed attention factor quantitatively computes the lower value concerning the patients' number in a medical crisis (emergency) and semantically computes the number of health specialties and the economic costs for the medical service. In addition, the geographic location ranks the medical facilities according to the user profile of the citizen, and the visualization is carried out in a Web-GIS application.

As collateral findings, we conclude that the efficient control of traffic congestion addressed the access to hospitals and facilities to provide health services according to geographic location and medical specialties. Therefore, it is a critical challenge that highly urbanized spaces such as Mexico City are facing. While such unexpected events are tough to avoid, novel computation and quick broadcast information about alternative routes could be the only way to decrease the loss of lives in health emergencies and services.

Thus, medical emergency responses require quick and reliable access and optimal routing. We know that road networks in megalopolises have become increasingly complex, and the density of traffic congestion is rising continuously, at least in Mexico City. Indeed, recommender systems in the healthcare context should be oriented towards improving

health services by incorporating filtering approaches to multi-criteria evaluation for health emergency routing services. Moreover, the new medical necessities require intelligent and automatic decision support considering the dynamic situation around the world.

On the other hand, the research limitations of the present approach are oriented toward the dependency on the Google Directions API service to obtain the estimations and approximations for the arrival time to the medical facilities. This occurs because the companies that manage huge data volumes based on Volunteered Geographic Information (VGI) do not share this information, and the data are not open and public. In this sense, our recommendation consists of developing frameworks based on VGI and crowd-sensing to generate massive data related to traffic congestion and other connected events. Concerning the semantic similarity measure, it is necessary to incorporate new measures to assess the information retrieval of the medical specialties, such as other semantic similarity measures but consider the high computational cost. Therefore, we suggest implementing the DIS-C, which is a conceptual distance measure proposed by Quintero et al. [49], to refine the semantic retrieval in the ontology.

Moreover, our future work will be focused on developing m-Health applications, considering the complete infrastructure of medical facilities, the establishment of mechanisms based on crowd-sensing and crowd-funding to evaluate massive and collaborative information concerning medical services, and clinical records of patients to generate prediction models based on machine learning methods, according to the healthcare services required by the citizens. At the same time, we are developing an approach based on ontology alignment to semantically interoperate with other medical ontologies, such as SNOMED, preserving the integrity of the conceptual representations. The goal is to produce more granularity in the medical specialties concerning information retrieval and offer new recommendations related to medical treatments, monitoring, and control of certain diseases.

**Author Contributions:** Conceptualization, R.Q. and M.T.-R.; methodology, G.G.; software, R.Q. and G.G.; validation, R.Q. and M.T.-R.; formal analysis, R.Q.; investigation, R.Q.; resources, M.T.-R.; data curation, G.G.; writing—original draft preparation, R.Q.; writing—review and editing, M.T.-R.; visualization, R.Q.; supervision, M.T.-R.; project administration, G.G.; funding acquisition, K.T.C. All authors have read and agreed to the published version of the manuscript.

**Funding:** This work was partially sponsored by the Instituto Politécnico Nacional and the Consejo Nacional de Ciencia y Tecnología under grants 20222091, 20220065, 20221777, and PN-2016/2110, respectively.

**Institutional Review Board Statement:** Not applicable.

**Informed Consent Statement:** Not applicable.

**Data Availability Statement:** Not applicable.

**Acknowledgments:** We are thankful to the reviewers for their time and their invaluable and constructive feedback that helped improve the quality of the paper.

**Conflicts of Interest:** The authors declare no conflict of interest.

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
