# Peer review of "Healthcare Recommender System Based on Medical Specialties, Patient Profiles, and Geospatial Information"

_sustainability, doi:10.3390/su15010499_

Round 1

Reviewer 1 Report

This paper is quite interesting, but there is some work for improving the quality of the paper.  

* The researcher(s) should pay attention to the research gap that is still not sufficient. Therefore, please add more arguments related to the research gap in the introduction.

*Please check line 50 "and the resilience of citizens. Indeed, 94" Also, please use the abbreviations once being introduced.  For example, FCA for Floating Catchment Area.

*The related work part was prepared very well. Yet, it's too long. Therefore, please try to restructure by only giving a short information.

*Please try to restructure the "conclusion and discussion" part, the research discussion should be in a separate section on the research conclusion. The current writing is not well structured.

*Along the same lines, it is necessary to mention the research limitations and recommendations in a separate section. *Some citation issues need to be fixed in the test. for example, in line 47 "According to [10]". The Ref. number is not allowed to be used as the subject in sentences of the main text (e.g., “[1] introduced …” is incorrect), I advise to add “Ref.” or “author names” before the citation, e.g., “[1] proposes a systematic” replace with “Ref. [1] proposes a systematic” or “Liu [1] proposes a systematic”. Consequently, please pay attention to the citations and references to check them appropriately.

*Finally, please improve the language of the research paper.   I hope that my comments can help you to improve your manuscript.

Author Response

Dear Reviewers and Editors,

We would like to thank you for the valuable time and advice from editors and reviewers to enhance the clarity and quality of our research work. Thus, we provide the answers to each comment suggested by the reviewers.

Reviewer 1

This paper is quite interesting, but there is some work for improving the quality of the paper. 

  1. *The researcher(s) should pay attention to the research gap that is still not sufficient. Therefore, please add more arguments related to the research gap in the introduction.

Answer: Thank you for the clarification. We have explained the research gap, including the issues and challenges of the recommender systems. It is presented in Section 1 “Introduction”, on page 2 from line 68 to line 106.

  1. *Please check line 50 "and the resilience of citizens. Indeed, 94" Also, please use the abbreviations once being introduced. For example, FCA for Floating Catchment Area.

Answer: Thank you for the comment. We have solved the problem of the abbreviations and completed the idea of “Indeed 94”, page 2, line 52.

  1. *The related work part was prepared very well. Yet, it's too long. Therefore, please try to restructure by only giving a short information.

 Answer: Thank you for the comment. We have reduced Section 2 “Related work” practically to two pages, and restructured the content with short and specific information. We have included a file with differences and track changes regarding the original submission and the new revised version.

  1. *Please try to restructure the "conclusion and discussion" part, the research discussion should be in a separate section on the research conclusion. The current writing is not well structured.

Answer: Thank you for the comment. We have taken into consideration your suggestion, and the discussion now is located in Section 4 “Results and discussion”. Additionally, we have described in more detail the discussion concerning the results. The new section for the conclusions in Section 5 “Conclusion and future work” was restructured and presented with more detail.

  1. *Along the same lines, it is necessary to mention the research limitations and recommendations in a separate section. *Some citation issues need to be fixed in the test. for example, in line 47 "According to [10]". The Ref. number is not allowed to be used as the subject in sentences of the main text (e.g., “[1] introduced …” is incorrect), I advise to add “Ref.” or “author names” before the citation, e.g., “[1] proposes a systematic” replace with “Ref. [1] proposes a systematic” or “Liu [1] proposes a systematic”. Consequently, please pay attention to the citations and references to check them appropriately.

Answer: Thank you for the comment. We have solved these issues in the entire document, and particularly in Section 2 “Related work” to put the appropriate citation for each reference.

  1. *Finally, please improve the language of the research paper. I hope that my comments can help you to improve your manuscript.

 Answer: Thank you for the comment. We have improved the language in the entire document. The changes can be appreciated in the file with differences and track changes.

Reviewer 2 Report

In this paper, the authors propose a novel, methodological, and collaborative approach based on patient-centered technology, which consists of a recommender system architecture to assist the health service level according to medical specialties. Extensively experimental results demonstrate that the efficiency of the proposed method. However, there are some issues that should be addressed.

1. The research motivation is unclear in Abstract, the authors should discuss the issues of recent methods on recommender system. 2. The contribution of this work is missing in Introduction. 3. Architecture and components of the recommender system in Figure 1 is too simple, which should present more descriptions. 4. The algorithm pseudocode of the proposed approach based on patient-centered technology is missing. 5. What is the loss function used by the authors, which they are trying to minimize? 6. The Figure 2, 3 seem a little blurry, which should be improved. 7. The experiments and tests in the section 4 are not convincing and need some improvements. It needs comparison against more existing models in terms of its validity. 8. The author maybe read these relevant literatures regarding algorithm to improve this paper. For instance, the following papers may be useful for your work, e.g., “A latent factor analysis-based approach to online sparse streaming feature selection”, “An L-and-L-Norm-Oriented Latent Factor Model for Recommender Systems”, “Symmetric nonnegative matrix factorization-based community detection models and their convergence analysis”, “Diversified Regularization Enhanced Training for Effective Manipulator Calibration”, “An Overview of Calibration Technology of Industrial Robots” and “A Novel Calibration System for Robot Arm via an Open Dataset and a Learning Perspective”.  

Author Response

Dear Reviewers and Editors,

We would like to thank you for the valuable time and advice from editors and reviewers to enhance the clarity and quality of our research work. Thus, we provide the answers to each comment suggested by the reviewers.

Reviewer 2

In this paper, the authors propose a novel, methodological, and collaborative approach based on patient-centered technology, which consists of a recommender system architecture to assist the health service level according to medical specialties. Extensively experimental results demonstrate that the efficiency of the proposed method. However, there are some issues that should be addressed.

  1. The research motivation is unclear in Abstract, the authors should discuss the issues of recent methods on recommender system.

 Answer: Thank you for the comment. We have improved the Abstract, highlighting the issues and motivation. The changes can be appreciated in the file with differences and track changes that were added in this revised version.

  1. The contribution of this work is missing in Introduction.

 Answer: Thank you for the comment. We have highlighted and emphasized the contribution of our research work. It is presented in Section 1 “Introduction”, on page 3 from line 110 to line 121.

  1. Architecture and components of the recommender system in Figure 1 is too simple, which should present more descriptions.

Answer: Thank you for the comment. We have redesigned Figure 1 and the processes or tasks are described now in the figure.

  1. The algorithm pseudocode of the proposed approach based on patient-centered technology is missing.

 Answer: Thank you for the comment. We have completed the algorithm of the proposed approach. It is depicted on page 11.

  1. What is the loss function used by the authors, which they are trying to minimize?

 Answer: Thank you for the comment. We have given a response regarding the loss function in the new revised version. It is presented on page 10 from line 363 to line 377.

  1. The Figure 2, 3 seem a little blurry, which should be improved.

Answer: Thank you for the comment. Now, Figure 2 is a vector; in this case, there is no blurry effect when making a zoom. Regarding Figure 3, it was modified with good resolution (vectorial format).

  1. The experiments and tests in the section 4 are not convincing and need some improvements. It needs comparison against more existing models in terms of its validity.

 Answer: Thank you for the comment. We have improved the experiments in Section 4. In addition, we included a new table (Table 3) in which a comparative process using Google Maps and Waze application in assessed with our proposed method. It is presented on page 14 from line 411 to line 426.

  1. The author maybe read these relevant literatures regarding algorithm to improve this paper. For instance, the following papers may be useful for your work, e.g., “A latent factor analysis-based approach to online sparse streaming feature selection”, “An L₁-and-L₂-Norm-Oriented Latent Factor Model for Recommender Systems”, “Symmetric nonnegative matrix factorization-based community detection models and their convergence analysis”, “Diversified Regularization Enhanced Training for Effective Manipulator Calibration”, “An Overview of Calibration Technology of Industrial Robots” and “A Novel Calibration System for Robot Arm via an Open Dataset and a Learning Perspective”.

 Answer: Thank you for the comment. We have reviewed and studied the recommended papers. Moreover, we thought that is very timely to cite some of these references (refs. 46, 47, 48) to clarify the loss function and the proposed algorithm.
